# Chemogenetic inhibition of the medial prefrontal cortex reverses the effects of REM sleep loss on sucrose consumption

Kristopher McEown, Yohko Takata, Yoan Cherasse, Nanae Nagata, Kosuke Aritake, Michael Lazarus*

International Institute for Integrative Sleep Medicine (WPI-IIIS), University of Tsukuba, Tsukuba, Japan

**Abstract** Rapid eye movement (REM) sleep loss is associated with increased consumption of weight-promoting foods. The prefrontal cortex (PFC) is thought to mediate reward anticipation. However, the precise role of the PFC in mediating reward responses to highly palatable foods (HPF) after REM sleep deprivation is unclear. We selectively reduced REM sleep in mice over a 25–48 hr period and chemogenetically inhibited the medial PFC (mPFC) by using an altered glutamate-gated and ivermectin-gated chloride channel that facilitated neuronal inhibition through hyperpolarizing infected neurons. HPF consumption was measured while the mPFC was inactivated and REM sleep loss was induced. We found that REM sleep loss increased HPF consumption compared to control animals. However, mPFC inactivation reversed the effect of REM sleep loss on sucrose consumption without affecting fat consumption. Our findings provide, for the first time, a causal link between REM sleep, mPFC function and HPF consumption.

*For correspondence: lazarus.
michael.ka@u.tsukuba.ac.jp

Competing interests: The
authors declare that no
competing interests exist.

Reviewing editor: Joel K
Elmquist, University of Texas
Southwestern Medical Center,
United States

## Introduction

A strong link exists between insufficient sleep and weight gain. A study of more than 30,000 participants found that those who slept less than 6 hr per night were more likely to gain weight over a 1-year period compared to people who slept 7 to 8 hr per night (*Watanabe et al., 2010*). Moreover, when healthy adults obtain insufficient sleep (i.e., 5 hr per night) they are more likely to consume weight-promoting foods compared to persons who obtain sufficient sleep (i.e., 9 hr per night; *Markwald et al., 2013*). Rapid eye movement (REM) sleep loss itself may be sufficient to increase consumption of weight promoting, highly palatable foods (HPF). *Shechter et al. (2012)* found that REM sleep loss in humans, over a period of 5 d, was inversely associated with hunger ratings and fat consumption. Indeed, increased intake of weight promoting, highly palatable foods may, over time, lead to adverse health outcomes such as obesity, diabetes and cardiovascular diseases.

Several studies have attempted to uncover the underlying neural mechanisms responsible for mediating an increased desire to consume HPF in response to sleep deprivation. Healthy adults who were subjected to one night of total sleep loss reported an increased desire to consume high-calorie foods compared to non-sleep deprived controls (*Greer et al., 2013*). In addition, sleep deprivation decreased activation, assessed by fMRI scans, in areas of the brain thought to mediate reward anticipation and inhibitory control, namely the anterior cingulate and orbitofrontal cortex, respectively (*Greer et al., 2013*; *Liu et al., 2011*; *Stuss, 2011*). However, the relationship between brain function and sleep loss are not consistent. For example, presentation of HPF to sleep deprived healthy adults was not associated with anterior cingulate activation (*St. Onge et al., 2014*) while in other instances a positive association was found between appetite for HPF and anterior cingulate function (*Benedict et al., 2012*).

Therefore, the current investigation sought to determine the role of the medial prefrontal cortex (medial PFC, mPFC) in mediating appetite for highly palatable foods in mice experiencing REM sleep loss. To achieve this aim, we used a well-validated chemogenetic method (*Lerchner et al., 2007*) to produce a modified invertebrate chloride channel (GluClαβ) that when activated by the drug ivermectin (IVM) facilitates neuronal inhibition in the mPFC over several days. We then examined the effect of mPFC inactivation and REM sleep loss on appetite for HPF in mice.

## Results

### Wire-mesh-grid device exposure produced REM sleep loss

During the dark period, mice exposed to a wire-mesh-grid device (WMGD) in their housing cage (*Figure 1A*) over a 72-hr period showed significantly less REM sleep after device introduction compared to baseline polygraphic recordings obtained without the device ($F_{3,6}$ = 23.835, p = 0.001; *Figure 1B and C*, *McEown et al., 2016*). There was a significant REM sleep reduction of 58% observed between baseline and the 25–48 hr dark time period (p = 0.026) and a 56% REM sleep reduction observed between baseline and the 49–72 hr dark time period (p = 0.020). In addition, a 46% REM sleep reduction was found when comparing the 0–24 hr and 25–48 hr dark time periods (p = 0.009). Moreover, we did not observe any rebound in REM sleep during the first 3 hr of the 25–48 hr light period (all comparisons p > 0.05; *Figure 1—figure supplement 1*). A significant increase in wakefulness was also produced by the device during the dark period ($F_{3,6}$ = 6.886, p = 0.023). Specifically, a 23% increase in wakefulness during the dark period was observed between baseline and the 49–72 hr time period (p = 0.042). No differences were observed between time periods, during the dark phase, for slow-wave sleep (SWS)—also known as non-REM sleep—($F_{3,6}$ = 4.605, p = 0.053). In addition, differences between time periods, during the light phase, for wake ($F_{3,6}$ = 1.608, p = 0.284), REM ($F_{3,6}$ = 1.103, 0.418) and SWS ($F_{3,6}$ = 1.434, p = 0.323) were also non-significant.

Exposing mice to the WMGD significantly affected sleep episode numbers during the 25–48 hr light and dark periods (*Figure 2A*). Specifically, WMGD exposure increased the number of SWS episodes during the light period, lasting 120 to 239 s, by 44% compared to baseline measures taken without the WMGD ($t_2$ = −18.898, p = 0.003). Furthermore, WMGD exposure decreased the number of REM episodes during the dark period, lasting 60 to 119 s, by 69% compared to baseline ($t_2$ = 9.430, p=0.011). On the other hand, WMGD exposure increased the number of wake episodes during the light period, lasting 30 to 59 s, by 42% compared to baseline ($t_2$ = −4.355, p = 0.049). The mean duration of SWS, REM sleep and wake episodes was not significantly different after WMGD exposure (*Figure 2B*).

SWS electroencephalogram (EEG) power density was also affected by WMGD exposure during the 25–48 hr light and dark periods (*Figure 2C*). That is, we observed increases in SWS EEG power density, of 12% to 14%, after exposure to the WMGD during the light period at 2.5 Hz ($t_2$ = −51.786, p = 0.000), 22 Hz ($t_2$ = −49.829, p = 0.000) and 24 Hz ($t_2$ = −31.982, p = 0.001) compared to baseline. We also observed increases in SWS EEG power density, of 12% to 14%, after WMGD exposure during the dark period at 2.5 Hz ($t_2$ = −42.819, p = 0.001), 3 Hz ($t_2$ = −27.978, p = 0.001), 10.5 Hz ($t_2$ = −121.636, p = 0.000) and 11 Hz ($t_2$ = −46.198, p = 0.000) compared to baseline. On the other hand, we did not observe significant differences in REM EEG power density during the 25–48 hr light or dark period (all comparisons p > 0.05; *Figure 2*, Panel C).

These results suggest that REM sleep is specifically reduced during the dark period in mice after WMGD exposure for the 25–48 hr time period, or longer, and thus the WMGD may be useful to elucidate the role of REM sleep loss on HPF consumption.

### GluClαβ-mPFC neuronal inhibition reverses the effect REM sleep loss on HPF consumption

Next, we investigated the effect of REM sleep loss on HPF consumption and whether changes in the latter one depend on the mPFC. Therefore, we used a chemogenetic system that involves a *Caeno-rhabditis elegans* glutamate-gated and IVM-gated chloride channel that was mutagenically modified to abolish sensitivity to glutamate while retaining sensitivity to IVM (*Lerchner et al., 2007*). The

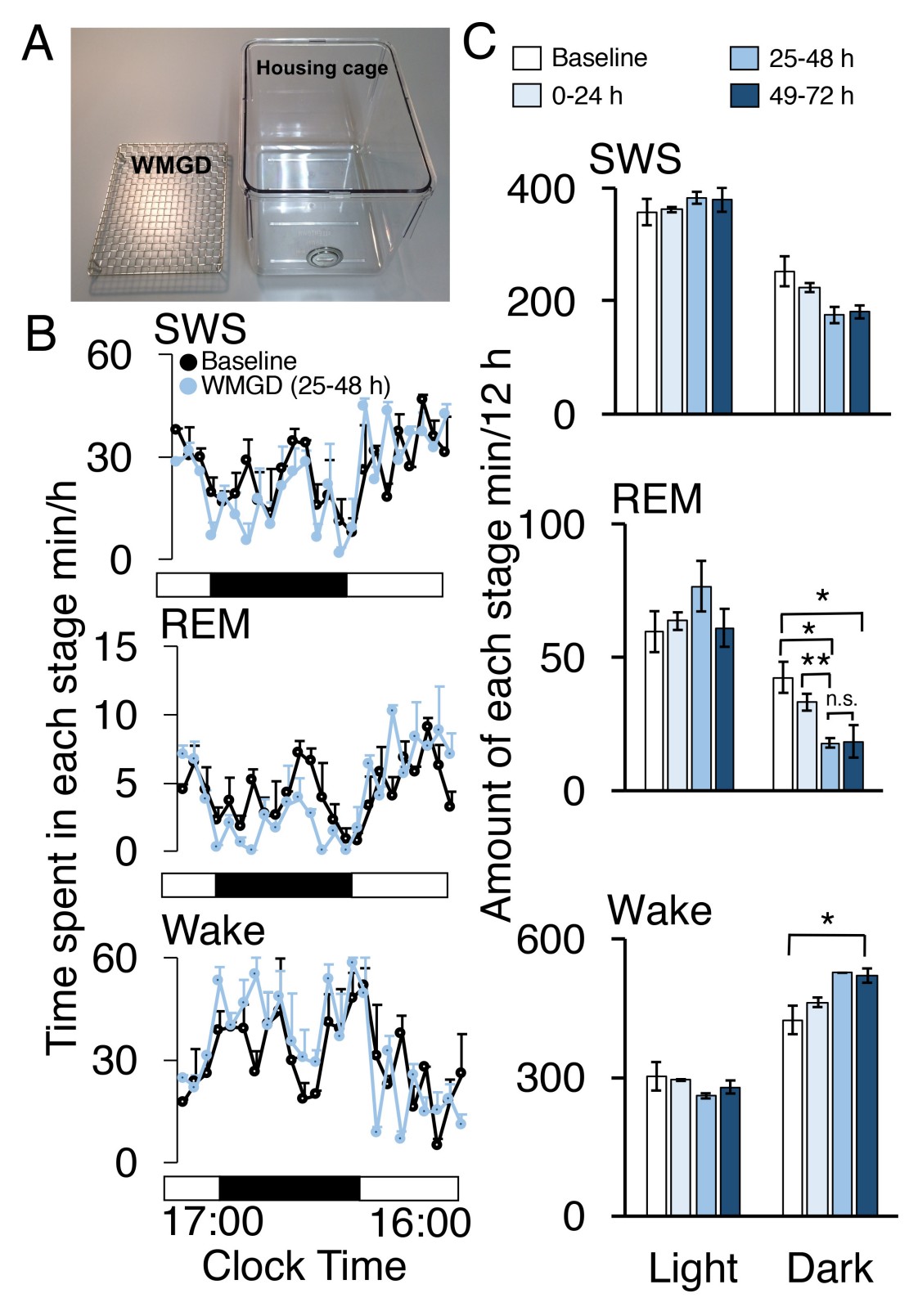

**Figure 1.** A wire-mesh-grid device (WMGD) produced REM sleep loss during the dark period in mice. (**A**) A pictorial representation of the WMGD (left) and the housing cage in which the WMGD was placed (right). (**B** and **C**) Time-courses (25–48 hr) of slow wave sleep (SWS), REM sleep and wake (**B**) and time spent in each sleep stage (**C**). Black and white bars in (**B**) indicate dark and light period, respectively. *p < 0.05 and **p < 0.01 indicate significant differences between baseline and WMGD exposure (n = 3).

*Figure 1 continued on next page*

*Figure 1 continued*

The following figure supplement is available for figure 1:

**Figure supplement 1.** REM sleep does not rebound during the light phase in mice exposed to the WMGD for 25–48 hr compared to baseline.

altered heteromeric channel, which is made up of an α-subunit and β-subunit, prevents the firing of action potentials by hyperpolarizing the cell membrane for several days after administration of IVM.

Two adeno-associated viruses (AAV) carrying the α-subunit or β-subunit of the IVM-gated GluClαβ channel (AAV-GluClα-IVM and AAV-GluClβ-IVM) were stereotaxically co-injected bilaterally into the mPFC of wild-type mice (*Figure 3A and B*). To detect expression of the GluClαβ channel, we labeled neurons with anti-green fluorescent protein (GFP) antibodies to stain the fusion proteins of the GluClα-subunit and GluClβ-subunit with yellow and cyan fluorescent proteins, respectively. Heat maps depicting AAV location and spread within the mPFC are displayed in *Figures 3B* and *4A*. The virus successfully targeted the mPFC, for mice used in all experiments, including the cingulate, prelimbic and infralimbic cortices (*Figure 3B*). IVM-injected mice had markedly attenuated Fos expression in the mPFC compared to mice injected with propylene glycol, indicating that the GluClαβ successfully inhibited neuronal activity in the mPFC (*Figure 3C*). SWS, REM and wakefulness amounts were non-significant when comparing the light and dark phases during the 25–48 hr time period for mPFC inactivated mice to the light and dark phases of the baseline day before the IVM injection (*Figure 3D*; all comparisons p>0.05).

We then examined the effect of REM sleep loss on sucrose consumption and whether changes in sucrose consumption depend on the mPFC ($F_{3,16} = 3.206$, p = 0.051; see *Figure 4*, Panel B). We compared non-sleep deprived mPFC inactivated mice to non-sleep deprived control mice and found no difference between these groups in sucrose (propylene glycol injected mice: p = 0.391; saline injected mice: p = 0.350) or fat consumption (propylene glycol injected mice: p = 0.575; saline injected mice: p = 0.434); indicating that mPFC inactivation, alone, did not affect sucrose or fat consumption in mice (*Figure 4—figure supplement 1*, Panel A). Moreover, we found that REM sleep loss increased sucrose consumption by 27% compared to non-sleep deprived saline injected controls (p = 0.039) and 35% compared to non-sleep deprived propylene glycol injected controls (p = 0.033). On the other hand, mPFC neuronal inhibition reversed the effect of REM sleep loss on sucrose consumption. Specifically, mPFC inactivated, REM sleep deprived mice consumed 49% less sucrose compared to REM sleep deprived controls (p = 0.004). Whereas sucrose consumption was non-significant between mPFC inactivated, REM sleep deprived mice and non-sleep deprived propylene glycol injected (p = 0.205) and saline injected controls (p=0.060).

Furthermore, the effect of REM sleep loss on fat consumption was also examined and whether changes in fat consumption depend on the mPFC ($F_{3,16} = 7.778$, p = 0.002; see *Figure 4*, Panel B). We found that REM sleep loss increased fat consumption by 26% compared to non-sleep deprived saline injected controls (p = 0.001) and 35% compared to non-sleep deprived propylene glycol injected controls (p = 0.000). However, mPFC inhibition paired with REM sleep loss did not affect fat consumption compared to sleep deprived controls (p = 0.199).

Baseline and follow-up weights of mice in the non-sleep deprived saline injected, non-sleep deprived propylene glycol injected, REM sleep reduced and mPFC inactivated groups were non-significant (Baseline: $F_{3,16} = 0.414$, p = 0.745; Follow-up: $F_{3,16} = 0.538$, p = 0.663). In addition, body weight did not significantly change between baseline and follow-up time points for the non-sleep deprived saline injected mice ($t_8 = 0.468$, p = 0.652), non-sleep deprived propylene glycol injected mice ($t_2 = 0.756$, p = 0.529), REM sleep reduced ($t_3 = 0.157$, p = 0.885) or PFC inactivated groups ($t_3 = 0.001$, p = 0.999). Finally, the amounts of sucrose (r = −0.175, p = 0.400) and fat (r = 0.246, p =0.240) consumed were not associated with baseline body weight (*Figure 4—figure supplement 1*, Panel B).

## Discussion

We were successfully able to mitigate the increased consumption of sucrose that resulted from REM sleep loss. We achieved this aim by chemogenetically inhibiting mPFC function. These results are the

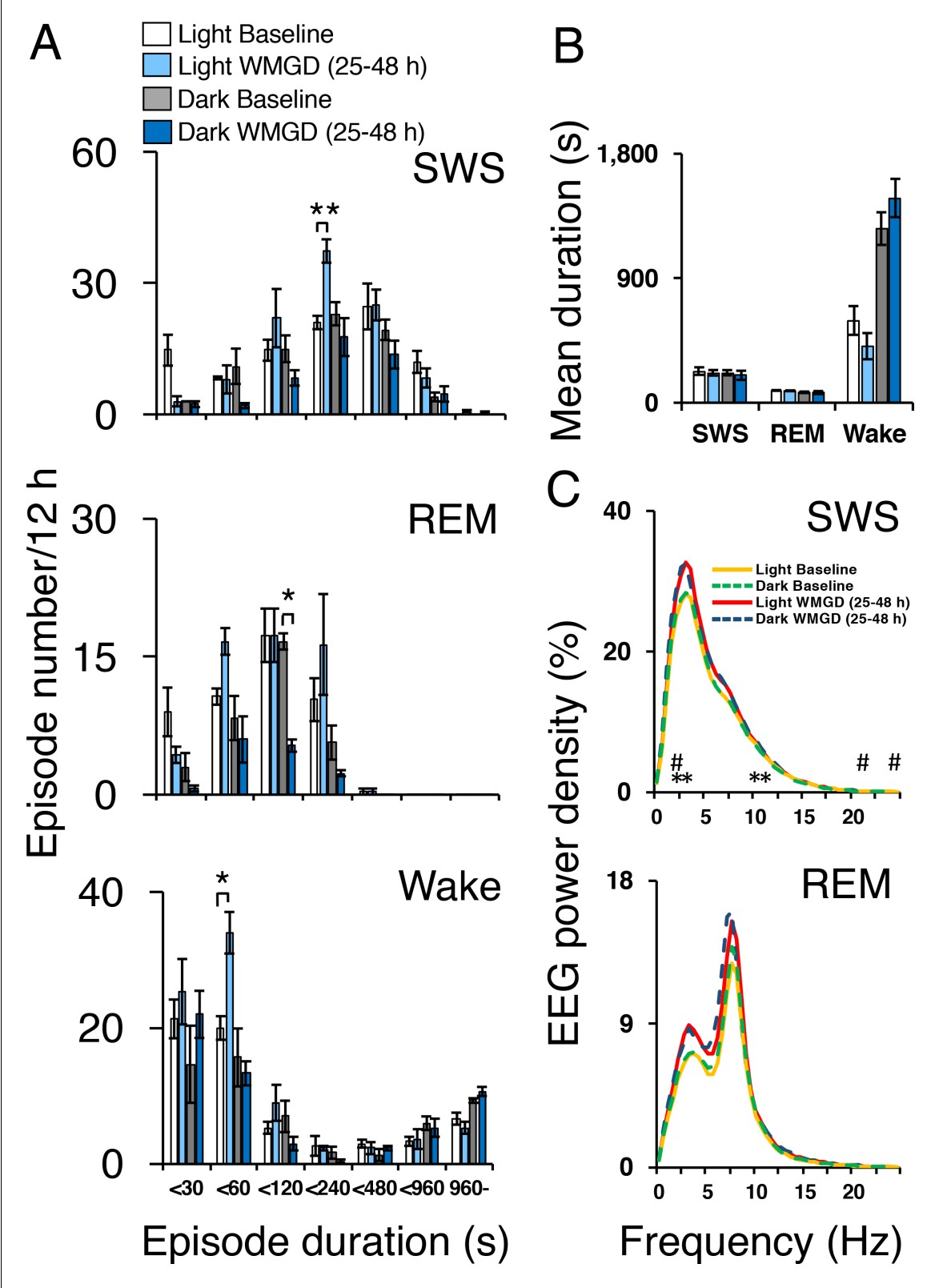

**Figure 2.** Sleep-wake profiles under baseline and wire-mesh-grid device (WMGD) conditions in mice for a 25–48 hr period, assessed by EEG/EMG recordings. (**A** and **B**) Effect of the WMGD on episode number (**A**) and duration (**B**) of slow wave sleep (SWS), REM sleep and wake during the light and dark phases. *p < 0.05 and **p < 0.01, indicates significant differences between baseline and WMGD exposure (n = 3). (**C**) EEG power density of SWS

*Figure 2 continued on next page*

Figure 2 continued

and REM sleep episodes in the light and dark phases. # and * symbols indicate significant differences (p < 0.01) between baseline and WMGD exposure during light (#) and dark phases (*).

first to demonstrate a clear, causative link between mPFC function, REM sleep and HPF consumption. Specifically, REM sleep loss increased consumption of two highly palatable foods, namely sucrose and fat. Whereas, mPFC inhibition completely reversed increased sucrose intake by attenuating the effect of REM sleep loss on sucrose consumption. On the other hand, mPFC inhibition did not attenuate the effect of REM sleep loss on fat consumption.

The PFC appears to play a role in mediating goal directed behavior toward highly appetitive food stimuli. Brain circuits responsible for judging the palatability of foods such as taste, texture and smell are mediated by areas of the PFC including the orbitofrontal and anterior cingulate cortex (*Rolls, 2015*). More specifically, the anterior cingulate appears to mediate taste for glucose in primates as feeding fruit juice (i.e., a high-glucose food) to satiety decreased neuronal firing in the anterior cingulate cortex using single cell recordings, whereas initial juice feeding increased neuronal firing in these animals (*Rolls, 2008*). Chocolate flavor also increases activation, as measured by fMRI, in the anterior cingulate cortex in humans (*Rolls and McCabe, 2007*). In addition, obese patients display increased PFC activity in response to high-calorie visual food stimuli; specifically, increased activity in the orbitofrontal and anterior cingulate cortex (*Stoeckel et al., 2008*; *Rothemund et al., 2007*). Alterations in PFC brain activity may, over time, result in the inability of these persons to inhibit salient food cues leading to excess high calorie food consumption and weight gain. Our results support these data by directly manipulating mPFC function in mice, which resulted in a substantial decrease in sucrose consumption, but had no effect on fat consumption. On the other hand, it is unknown whether consumption of other highly palatable foods may be effected by mPFC inactivation and/or REM disruption (e.g., foods high in salt content). Future research is needed to explore this possibility. The mPFC contains heterogeneous neurons that include some inhibitory GABAergic interneurons (approximately 20%) and excitatory pyramidal neurons (approximately 80%; (*Riga et al., 2014*). Moreover, mPFC neurons expressing dopamine D1 receptors mediate food consumption (*Land et al., 2014*); however, it is currently unknown what role this neuronal subtype may play in controlling highly palatable food consumption in response to REM sleep disruption.

Furthermore, in humans REM sleep loss adversely affects dietary behavior which may lead to weight gain. For example, REM sleep loss in humans enhances the desire to consume food and increases food consumption (*Gonnissen et al., 2013*; *Shechter et al., 2012*). In addition, REM sleep loss in children and adolescents, of 1 hr per night measured over three nights, was associated with a marked increase in the likelihood of these children being overweight (*Liu et al., 2008*). In agreement with these findings, we observed that reducing REM sleep, over a 25–48 hr period, during the dark phase resulted in increased HPF consumption. On the other hand, it should be noted that, in the current investigation, REM sleep was not effected during the light phase. It is unknown why our observed reduction in REM sleep was restricted to the dark phase and not during the light phase. Indeed, sleep pressure is greater during the light phase compared to the dark phase in mice; therefore, it is possible that our device was only able to produce REM reduction during the dark phase when sleep pressure was weaker. In addition, the REM sleep disruption observed in the current investigation did not occur during the first 24 hr after WMGD introduction. A plausible explanation is that the sleep reducing effect of our device requires time to build-up and thus produce disturbances in sleep.

In conclusion, our results demonstrated that mPFC inhibition reduced sucrose consumption in REM sleep deprived mice. Our results clearly indicate that increased HPF consumption, resulting from REM sleep loss, is mediated by mPFC function thereby providing clear findings that elucidate the role of the mPFC and REM sleep loss in mediating unhealthy dietary behavior, which may lead to adverse health outcomes.

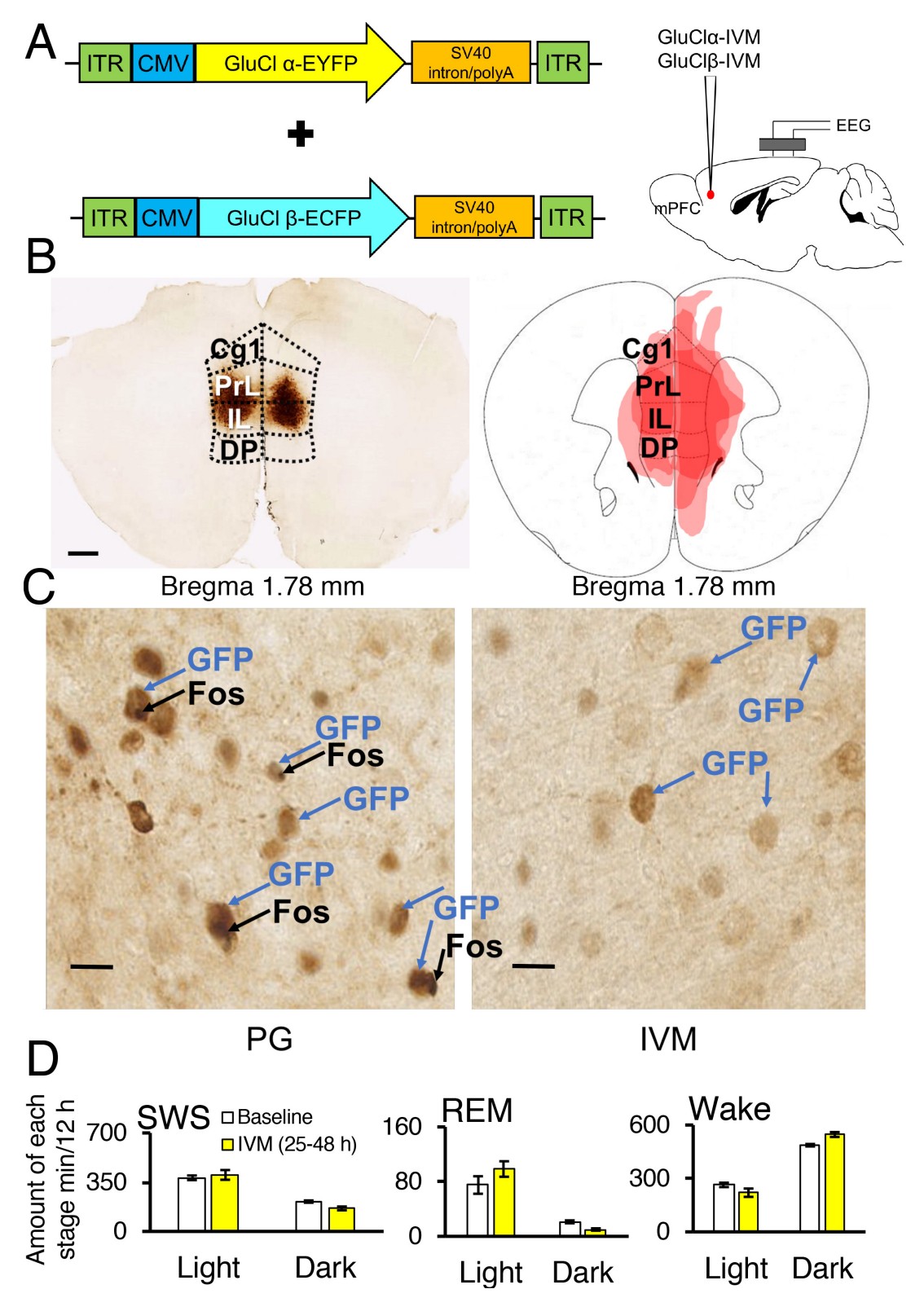

**Figure 3.** Chemogenetic inhibition of medial prefrontal cortex (mPFC) neurons in mice. (**A**) Wild-type mice were injected with GluClα-IVM and GluClβ-IVM adeno-associated viruses (AAV; left) into the mPFC and implanted with somnographic electrodes (right). (**B**) Brain sections were stained against fluorescent protein (GFP) to confirm that GluClα and GluClβ proteins were expressed in the mPFC. Scale bar: 500 μm (left). Drawings of superimposed AAV injection sites in the mPFC are shown on the right (n = 3). (**C**) GFP (brown) and Fos (black) expression in two mice injected with AAV-GluClα-IVM

*Figure 3 continued on next page*

*Figure 3 continued*

and AAV-GluClβ-IVM into the mPFC and treated with propylene glycol (PG) or ivermectin (IVM). Scale bars: 20 µm. Blue and black arrows indicate neurons expressing GFP and Fos, respectively. (D) Time spent in slow wave sleep (SWS), REM sleep and wake between 25–48 hr after IVM treatment in mice injected with AAV-GluClα-IVM/AAV-GluClβ-IVM into the mPFC. Other abbreviations: ITR, inverted terminal repeat; CMV, cytomegalovirus; SV40, simian virus 40; EYFP, enhanced yellow fluorescent protein; ECFP, enhanced cyan fluorescent protein; Cg1, cingulate cortex; PrL, prelimbic cortex; IL, infralimbic cortex; DP, dorsal peduncular cortex.

## Materials and methods

### Subjects and chemicals

A male mouse line on a C57BL/6 background, which was maintained at the International Institute of Integrative Sleep Medicine and weighed between 23 and 34 g (9–10 weeks old) at the beginning of the experiments, was used for all experiments. The animals were individually housed in insulated sound-proof chambers maintained at an ambient room temperature of 22°C ± 1°C with a relative humidity of 60 ± 2% in an automatically controlled 12 hr light/dark cycle (lights on at 8:00, lights off at 20:00). Food and water were available *ad libitum*. No method of blinding or randomization was used for any of the experiments. This study was performed in strict accordance with the recommendations in the Guide for the Care and Use of Laboratory Animals of the National Institutes of Health. All animals were handled according to approved institutional animal care and use committee (IACUC) protocols (#16–359) of the University of Tsukuba. All protocols aimed to reduce the number of animals used for each experiment and to minimize pain or discomfort. IVM was obtained from Merial Company.

### AAV vector generation

The AAV of serotype rh10 for AAV10-α-GluCl-IVM-EYFP (AAV-GluClα-IVM) and AAV10-β-GluCl-IVM-ECFP (AAV-GluClβ-IVM) were generated by tripartite transfection (AAV-rep2/caprh10 expression plasmid, adenovirus helper plasmid, and pAAV-GluClα-IVM or pAAV-GluClβ-IVM plasmid) into 293A cells. After 3 days, 293A cells were resuspended in artificial cerebrospinal fluid (aCSF), freeze-thawed four times, and treated with benzonase nuclease (Millipore) to degrade all forms of DNA and RNA. Subsequently, the cell debris was removed by centrifugation and the virus titre in the supernatant was determined using an AAVpro Titration Kit for Real Time PCR (Takara) (See *Figure 3*, Panel A).

### Stereotaxic AAV injection and placement of EEG/EMG electrodes

Surgeries for AAV injections were conducted under pentobarbital anaesthesia (50 mg/kg, intraperitoneal [i.p.]). Using aseptic techniques, mice were injected stereotaxically into the mPFC (n = 14) with recombinant GluClα-AAV and GluClβ-AAV (250 nl/injection, $1.1 \times 10^{11}$ particles/ml), with a glass micropipette and an air pressure injector system (*Chamberlin et al., 1998*). The following coordinates were used for injections into the mPFC of C57BL/6 mice according to the atlas of *Paxinos and Franklin (2001)*. Bilateral injections were made at 1.7 mm anterior and 0.4 mm lateral to bregma, and 2.0 mm below the dural surface.

Mice were also chronically implanted with EEG and electromyogram (EMG) electrodes for polysomnography (n = 6) as previously outlined (*Oishi et al., 2016*). Briefly, the implant comprised two stainless steel screws (1 mm diameter) serving as EEG electrodes, one of which was placed epidurally over the right frontal cortex (1.0 mm anterior and 1.5 mm lateral to bregma) and the other over the right parietal cortex (1.0 mm anterior and 1.5 mm lateral to lambda). Two insulated Teflon-coated, silver wires (0.2 mm in diameter), which were placed bilaterally into the trapezius muscles, served as EMG electrodes. Both EEG and EMG electrodes were connected to a microconnector, and the whole assembly was then fixed to the skull with self-curing dental acrylic resin. Animals were allowed one week for postoperative recovery before being placed in experimental housing cages (*Figure 1*, Panel A) for a 24 hr habituation period and connected with recording leads.

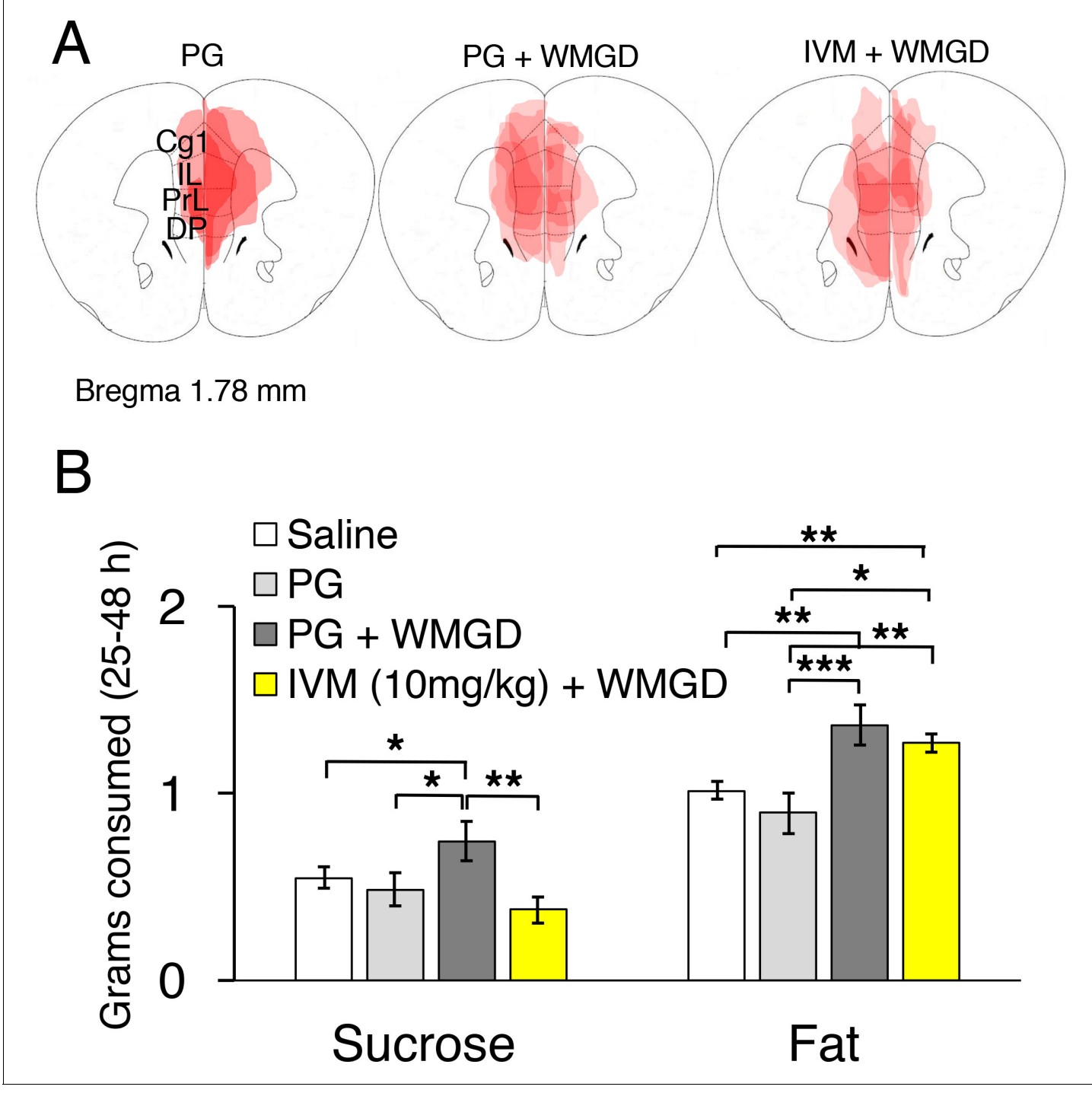

**Figure 4.** Chemogenetic inhibition of medial prefrontal cortex (mPFC) neurons reverses the effect of REM sleep loss on highly-palatable-foods consumption. (A) Drawings of superimposed adeno-associated viruses (AAV) injection sites in the mPFC of mice used for baseline (n = 3; left panel) or wire-mesh-grid device (WMGD) experiments [n = 4 in the propylene glycol (PG), middle panel, or ivermectin (IVM), right panel, treated group]. (B) Grams of sucrose or fat consumed over a 25–48 hr period under baseline (nine wild-type mice treated with saline and three AAV-injected mice treated with PG) and WMGD [AAV-injected mice treated with PG (n = 4) or IVM (n = 4)] conditions. *p < 0.05, **p < 0.01 and ***p < 0.001 indicates significant differences between mice groups (one-tailed).

The following figure supplement is available for figure 4:

**Figure supplement 1.** Control experiments for highly palatable foods consumption in mice.

## EEG/EMG recording

After the 24-hr habituation period, mice were injected with vehicle (propylene glycol, i.p.) at 16:30. Baseline polygraphic recording commenced at 17:00 and terminated at 16:00 the following day. After baseline recordings were obtained, mice (n = 3) were injected with IVM (10 mg/kg; i.p.) at 16:30 on the following day to initiate virus activation. Polygraphic recordings subsequently commenced at 17:00 for a period of 72 hr.

A separate group of EEG/EMG implanted mice (n = 3) were tested using the preceding methods with the exception that mice did not receive AAV infusions and did not receive i.p. injections. Instead, a wire-mesh-grid cage bottom (purchased from Oriental Yeast Company, Tokyo, Japan) was introduced, at 16:30 the day after baseline polygraphic recording, for a period of 72 hr. The wire-mesh-grid cage bottom was constructed from stainless steel and measured: 22 cm (length) × 19.5 cm (width) × 2.5 cm (height) with the grid consisting of 1 cm$^2$ openings (see *Figure 1*, Panel A).

## Vigilance state assessment based on EEG/EMG/locomotor activity recordings

The EEG/EMG signals were amplified and filtered (EEG: 0.5–64 Hz, EMG: 16–64 Hz), then digitised at a sampling rate of 128 Hz, and recorded using SLEEPSIGN software (*Kohtoh et al., 2008*) (Kissei Comtec). In addition, locomotor activity was recorded with an infrared photocell sensor (Biotex). The vigilance states were scored offline by 10 s epochs into three stages, including waking, SWS, and REM sleep, according to standard criteria (*Oishi et al., 2016*). As a final step, defined vigilance stages were examined visually, and corrected when necessary.

## HPF data collection

HPF consumption measurements were obtained from mice that did not receive EEG/EMG electrode implantation (n = 24). After AAV infusion, mice were singly placed into testing cages without the wire-mesh-grid-cage bottoms for 7 days. During these seven days mice had free access to laboratory chow. On day 8, mice were injected with IVM (10 mg/kg, i.p.; n = 4) or vehicle (propylene glycol; n = 4) at 16:30 and singly placed back into their housing cages with the addition of the wire-mesh-grid cage bottoms at 17:00 (*Figure 1*, Panel A). The highly palatable foods were introduced on day 8 and food consumption was subsequently measured every 24 hr at 17:00 over a total period of 72 hr. In addition, three controls groups were used in which one group of mice did not receive AAV infusions and were injected with saline (n = 9), one group of mice received AAV infusions and were injected with propylene glycol (n = 3), and one group of mice received AAV infusions and were injected with IVM (n = 4). All three groups were not exposed to the wire-mesh-grid cage bottom (i. e., the non-sleep deprived groups). During the 72 hr test period, all mice had free access to white chocolate (DARS brand, Morinaga Company, Tokyo, Japan), a high-fat diet (HFD-60, Oriental Yeast Company, Tokyo, Japan) and standard laboratory chow (see *Supplementary file 1* for the composition of each highly palatable diet). Total amounts of sucrose and fat consumption were calculated from each food type (i.e., sucrose and fat amounts from the white chocolate and from the high-fat diet) 25–48 hr after initial food exposure.

## Immunohistochemistry for GFP and Fos immunoreactive cell expression

When data collection, outlined above, was complete mice were deeply anesthetized using chloral hydrate (500 mg/kg, i.p.) and perfused through the left ventricle of the heart with saline and subsequently perfused with neutral buffered 10% formalin solution. Brains were then removed and placed into vials containing 10% formalin solution for 1 week. Brains were subsequently transferred to vials containing 20% sucrose in phosphate-buffered saline (PBS) for 24 hr at 4°C to reduce freezing artifacts. Brains were then frozen using dry ice, placed on a freezing microtome and sectioned at 40 μm. Immunohistochemistry was undertaken on free floating brain sections as previously described (*Lazarus et al., 2011*). In brief, sections were rinsed in PBS, incubated in 0.3% hydrogen peroxide in 0.25% Triton X-100 in PBS (PBT) for 30 min at room temperature, rinsed in PBS and incubated, overnight at room temperature, in rabbit GFP primary antibody (Molecular Probes, RRID:AB_221569, lot number 1293114) at a dilution of 1:20,000 or rabbit Fos primary antibody (Millipore, RRID:AB_2106755, lot number D00058535) at a dilution of 1:20,000 in PBT with 0.02% sodium azide. After overnight incubation, brain sections were rinsed in PBS for 1 hr and then incubated for 90 min in

biotinylated anti-rabbit antibody (Jackson ImmunoResearch) at a dilution of 1:1000. These sections were then treated with avidin-biotin complex (1:1000; Vectastain ABC Elite kit, Vector Labs) for 1 hr and immunoreactive cells were visualised by reaction with 3,3'-diaminobenzidine and 0.01% hydrogen peroxide. Tissue sections mounted on glass slides were scanned with a Hamamatsu Nano-Zoomer-XR Digital slide scanner (Hamamatsu Photonics), and digital photomicrographs were analyzed with Hamamatsu NDPView software v2.4.26.

AAV location was assessed by visualizing GFP expression in brain sections using Hamamatsu NDPView software v2.4.26. Virus location figures (see *Figure 3*, Panel B and *Figure 4*, Panel A) were produced by using the following procedure: (a) observing virus location and spread within brain sections; (b) transforming digital photomicrographs images to color-scale images; and (c) superimposing these color-scale images to obtain a summation of AAV location for all mice.

Furthermore, Fos immunoreactive cell expression was measured using two AAV infused mice that were injected with either propylene glycol or IVM, exposed to the WMGD for 48 hr and fed a diet of white chocolate, HFD and chow. After WMGD exposure, mice were sacrificed during the dark period (i.e. at 21:00). Immunohistochemistry procedures were then undertaken as outlined above. Fos immunoreactive cell imaging, at 20 times magnification, was produced using Hamamatsu NDPView software v2.4.26 (see Panel C, *Figure 3*) by focusing on an area of interest located within the mPFC at 1.7 mm anterior and 0.4 mm lateral to bregma, and 2.0 mm below the dural surface.

## Statistical analyses

Means and ± S.E.M. are expressed for all statistical comparisons. For comparisons between two groups, two-tailed paired samples Student's t-tests were used followed by Bonferroni corrections. In addition, Pearson correlations were used to compare associations between two groups. Repeated measures ANOVA or one-way ANOVA were used when comparing groups of three or more followed by Fisher's Probable Least-Squares Difference (PLSD) post hoc tests. For comparisons between independent samples, the Levene's test was used to assess homogeneity of variance and for paired sample comparisons the Mauchly's test was used to assess sphericity. All comparisons were considered statistically significant at $p < 0.05$. Statistics for all data are reported in *Supplementary file 2*.

## Acknowledgements

This work was supported by a Japan Society for the Promotion of Science KAKENHI grant (2604762, to ML) and postdoctoral fellowship (P14762, to KM); a Takeda Science Foundation postdoctoral fellowship (to KM); a grant from the Ministry of Education, Culture, Sports, Science and Technology (MEXT) of Japan (Grant-in-Aid for Scientific Research on Innovative Areas 'Living in Space', 16H01629, to ML); a CREST grant from the Japan Science and Technology Agency (to ML) and the World Premier International Research Center Initiative (WPI) from MEXT (to ML, YT, YC, NN and KA).

## Additional information

### Funding

| Funder | Grant reference number | Author |
|---|---|---|
| Japan Society for the Promotion of Science | P14762 | Kristopher McEown |
| Japan Society for the Promotion of Science | 2604762 | Michael Lazarus |
| Ministry of Education, Culture, Sports, Science, and Technology | 16H01629 | Michael Lazarus |
| Japan Science and Technology Agency | 16814578 | Michael Lazarus |

The funders had no role in study design, data collection and interpretation, or the decision to submit the work for publication.

## Author contributions

KM, Conception and design, Acquisition of data, Analysis and interpretation of data, Drafting or revising the article; YT, NN, Acquisition of data, Drafting or revising the article; YC, Drafting or revising the article, Contributed unpublished essential data or reagents; KA, Analysis and interpretation of data, Drafting or revising the article; ML, Conception and design, Analysis and interpretation of data, Drafting or revising the article

## Ethics

Animal experimentation: This study was performed in strict accordance with the recommendations in the Guide for the Care and Use of Laboratory Animals of the National Institutes of Health. All of the animals were handled according to approved institutional animal care and use committee (IACUC) protocols (#16-359) of the University of Tsukuba. All protocols aimed to reduce the number of animals used for each experiment and to minimize pain or discomfort.

# Additional files

## Supplementary files

• Supplementary file 1. Composition of the highly palatable diets used in the feeding experiments.

• Supplementary file 2. Statistics reported in this study.

## Major datasets

The following dataset was generated:

| Author(s) | Year | Dataset title | Dataset URL | Database, license, and accessibility information |
|---|---|---|---|---|
| McEown K, Takata Y, Cherasse Y, Nagata N, Aritake K, Lazarus M | 2016 | Data from: Chemogenetic inhibition of the medial prefrontal cortex reverses the effects of REM sleep loss on sucrose consumption | http://dx.doi.org/10.5061/dryad.4v58b | Available at Dryad Digital Repository under a CC0 Public Domain Dedication |

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
