## [Decision Letter]

Thank you for submitting your article "Chemogenetic inhibition of the medial prefrontal cortex reverses the effects of REM sleep loss on sucrose consumption" for consideration by *eLife*. Your article has been favorably evaluated by Timothy Behrens (Senior Editor) and three reviewers, one of whom is a member of our Board of Reviewing Editors.

The reviewers have discussed the reviews with one another and the Reviewing Editor has drafted this decision to help you prepare a revised submission.

Summary:

McEown and colleagues describe results from a series of studies investigating the role of neurons in the medial prefrontal cortex (mPFC) in regulating and linking REM sleep to consumption of highly palatable food. They demonstrate that sleep loss induces an increase of a diet rich in sucrose. Notably, chemogenetic inhibition of mPFC neurons blunted that increase. They did not find an effect to blunt a high fat diet. Collectively, the results are novel and are potentially of wide interest. It is widely accepted that sleep disruption affects metabolism and energy balance. However, the CNS sites and mechanisms linking sleep disruption and obesity are poorly defined. However, as it currently stands the manuscript has several issues that need to be addressed.

Essential revisions:

The feeding studies are not well described and at this point seem somewhat preliminary. It is not clear if the mice were singly housed and how long they were acclimated to the diets and conditions. Also, it would be useful to know if the REM disruption paradigm affects body weight and composition over time. If this has been published it needs to be pointed out.

The authors allude to the fact that the mPFC neurons are affecting taste/palatability. Have previous studies investigated the role of these neurons in regulating taste?

Is this effect really specific for sucrose? By only having one comparison it is hard to judge the behavioral specificity of the PFC action.

The control of the effect of inactivation of the medial prefrontal cortex in basal condition on food consumption is missing. Indeed, it is important to verify whether the medial prefrontal cortex has also an effect in control condition. One question raised by the results is indeed whether the results are specific to the REM condition. It might be that the medial prefrontal cortex is indeed needed for sucrose consumption.

What is the difference between loss of REM in dark v. light period? Since the WMGD has the largest effect in the dark period, when mice are normally awake, the exact role of REM loss is a bit ambiguous. Some further clarification for how this works and that the mice do not catch up the light period would be helpful.

The authors used an original way to deprive mice of REM sleep. They put a wire mesh grid on the bottom of the mice cages. By this means, they obtained a significant decrease of REM sleep quantities during nighttime but not during the day. They also induced a small increase in wake quantities and a non-significant decrease in SWS quantities. I'm wondering why they did not see an effect in the first 24h? This is not mentioned in the manuscript. They should also discuss why there is such a delayed onset of the effect? Also, how they explained it occurs only during the night?

---

## [Author Response]

*[…] Essential revisions:*

*The feeding studies are not well described and at this point seem somewhat preliminary. It is not clear if the mice were singly housed and how long they were acclimated to the diets and conditions. Also, it would be useful to know if the REM disruption paradigm affects body weight and composition over time. If this has been published it needs to be pointed out.*

We have added further detail to the feeding studies methods to better describe our experiments (please see subsection “HPF data collection”). Mice were singly housed and this point is now stated in the aforementioned subsection. Subjects were acclimated to the housing conditions for one week prior to introduction of the highly palatable diet or the wire-mesh-grid-device. Mice were not acclimated to the highly palatable diets prior to commencing the feeding studies and the REM disruption did not affect body weight over our 72 h testing period (please see subsection “GluClαβ-mPFC neuronal inhibition reverses the effect REM sleep loss on HPF consumption”, last paragraph). We did not measure body composition over this 72 h period, however it is unlikely that our intervention would affect body composition (e.g., muscle, fat, etc.) over this short time period. We have not examined whether REM disruption, using our device, would affect body weight or composition over longer periods of time.

*The authors allude to the fact that the mPFC neurons are affecting taste/palatability. Have previous studies investigated the role of these neurons in regulating taste?*

Previous studies have investigated the role of mPFC neurons in regulating taste/palatability. We have added a few examples of this research as follows:

“More specifically, the anterior cingulate appears to mediate taste for glucose in primates as feeding fruit juice (i.e., a food high in glucose) to satiety decreased neuronal firing in the anterior cingulate cortex using single cell recordings, whereas initial juice feeding increased neuronal firing in these animals (Rolls et al. 2008). Chocolate flavor also increases activation, as measured by fMRI, in the anterior cingulate cortex in humans (Rolls and McCabe 2007).”

Is this effect really specific for sucrose? By only having one comparison it is hard to judge the behavioral specificity of the PFC action.

The mPFC may mediate the taste/palatability for other foods. Therefore, the effect of REM disruption may not be specific to only sucrose. Future research is needed to address this possibility. We have added sentences to the Discussion to address this possibility as follows:

“On the other hand, it is unknown whether consumption of other highly palatable foods may be effected by mPFC inactivation and/or REM disruption (e.g., foods high in salt content). Future research is needed to explore this possibility.”

*The control of the effect of inactivation of the medial prefrontal cortex in basal condition on food consumption is missing. Indeed, it is important to verify whether the medial prefrontal cortex has also an effect in control condition. One question raised by the results is indeed whether the results are specific to the REM condition. It might be that the medial prefrontal cortex is indeed needed for sucrose consumption.*

We have now added these data to the text (subsection “GluClαβ-mPFC neuronal inhibition reverses the effect REM sleep loss on HPF consumption”, third paragraph) and in a supplement to Figure 4 (i.e., Figure 4—figure supplement 1, Panel A). Our data show that the medial prefrontal cortex did not mediate sucrose consumption as non-sleep deprived medial prefrontal cortex inactivated mice did not differ in the amount of sucrose consumed compared to non-sleep deprived control mice.

*What is the difference between loss of REM in dark v. light period? Since the WMGD has the largest effect in the dark period, when mice are normally awake, the exact role of REM loss is a bit ambiguous. Some further clarification for how this works and that the mice do not catch up the light period would be helpful.*

It is unknown why REM disruption was confined to the dark period in our study. We have addressed these points as follows:

“It is unknown why our observed reduction in REM sleep was restricted to the dark phase and not during the light phase. Indeed, sleep pressure is greater during the light phase compared to the dark phase in mice, therefore it is possible that our device was only able to produce REM reduction during the dark phase when sleep pressure was weaker.”

We have added a supplement to Figure 1 (i.e., Figure 1—figure supplement 1) showing that REM sleep does not rebound during the first three hours of the light period 25-48 h after WMGD introduction. We have also addressed this in text (subsection “Wire-mesh-grid device exposure produced REM sleep loss”, first paragraph).

*The authors used an original way to deprive mice of REM sleep. They put a wire mesh grid on the bottom of the mice cages. By this means, they obtained a significant decrease of REM sleep quantities during nighttime but not during the day. They also induced a small increase in wake quantities and a non-significant decrease in SWS quantities. I'm wondering why they did not see an effect in the first 24h? This is not mentioned in the manuscript. They should also discuss why there is such a delayed onset of the effect? Also, how they explained it occurs only during the night?*

Indeed, we do not know why our device successfully reduced REM sleep during the dark period but not during the light period. However, we have addressed this point in the third paragraph of the Discussion. In addition, we have addressed the delayed onset of the REM effect as follows:

“In addition, the REM sleep disruption observed in the current investigation did not occur during the first 24 h after WMGD introduction. A plausible explanation is that the sleep reducing effect of our device requires time to build-up and thus produce disturbances in sleep.”